# LOXL4 Abrogation Does Not Exaggerate Angiotensin II-Induced Thoracic or Abdominal Aortic Aneurysm in Mice

**DOI:** 10.3390/genes12040513

**Published:** 2021-03-31

**Authors:** Huimin Li, Jun Guo, Yiting Jia, Wei Kong, Wei Li

**Affiliations:** 1Beijing Key Laboratory for Genetics of Birth Defects, Beijing Pediatric Research Institute, MOE Key Laboratory of Major Diseases in Children, Capital Medical University, Center of Rare Diseases, National Center for Children’s Health, Beijing Children’s Hospital, Capital Medical University, Beijing 100045, China; lihuiminwin@126.com (H.L.); guojunbjmu@hotmail.com (J.G.); 2Department of Physiology and Pathophysiology, School of Basic Medical Sciences, Peking University, Beijing 100019, China; yitingjia@bjmu.edu.cn (Y.J.); kongw@bjmu.edu.cn (W.K.)

**Keywords:** angiotensin II, LOXL4, thoracic or abdominal aortic aneurysm, variants, whole-exome sequencing

## Abstract

It has been shown that thoracic aortic aneurysm and dissection (TAAD) could be a Mendelian trait caused by a single gene mutation. The *LOX* gene mutation leads to the development of human TAAD. The *LOXL4* gene is a member of the lysyl oxidase gene family. We identified seven variants in the *LOXL4* gene in 219 unrelated patients with TAAD by whole-exome sequencing (WES). To further investigate whether *LOXL4* is a candidate causative gene for human TAAD, a *Loxl4* knockout mouse was generated, and the mutant mice were treated by subcutaneous infusion of angiotensin II. We found that abrogation of *Loxl4* did not induce a more severe thoracic or abdominal aortic aneurysm compared with the wild-type C57BL/6J mice. Our results suggest that LOXL4 may not play a major role in the development of angiotensin II-induced aortic aneurysm. The functional study using this animal model system is important for the evaluation of candidate genes of TAAD identified by WES.

## 1. Introduction

Aortic aneurysm (AA) is a degenerative condition that leads to vascular remodeling and luminal expansion of the aorta wall. When the arterial wall cannot withstand the internal blood pressure, aortic dissection or rupture will happen, which is often asymptomatic and life-threatening. According to the location, aortic aneurysms are divided into thoracic aortic aneurysms (TAA), abdominal aortic aneurysms (AAA), and thoracic-abdominal combined aneurysms. The annual incidence of TAA is about 7~9/100,000 [1]. The incidence of AAA increases with age, up to 8.8% among 65-year-old people [2].

Elastin and collagen are two major structural components of the arterial wall. Lysyl oxidase (LOX) and its related gene family members LOXL1~4 (LOX-like proteins 1~4) are a group of extracellular copper enzymes that catalyze the formation of lysine-derived crosslinks in elastin and hydroxylysine-derived crosslinks in collagens. It has been reported that all LOX family members are expressed in the developing aorta, and the *LOX* gene is the major LOX family member [3]. Variants of the *LOX* gene disrupt enzyme function and predispose to thoracic aortic aneurysm and dissection (TAAD) in humans [4,5]. However, the relationship between *LOXL1~LOXL4* and TAAD has not been defined. In our previous study, 219 sporadic patients with TAAD were studied by whole-exome sequencing (WES), and some results have been published [6]. To illustrate whether genes encoding LOX-like proteins were causative genes of TAAD, we analyzed variants of *LOXL1*, *LOXL2*, *LOXL3,* and *LOXL4* in 219 patients with TAAD. Seven variants of *LOXL4* were identified in eight patients. LOXL4, as a downstream molecule of TGF-β signaling in aortic endothelial cells, participates in the deposition and assembly of aortic extracellular matrix through extracellular secretion [7]. This highlights that *LOXL4* could be a candidate gene of TAAD. There is no report on *Loxl4* knockout in any animal model, and its function in vivo is unclear. We generated a *Loxl4* knockout mouse to investigate whether *Loxl4* abrogation could exaggerate the development of TAAD.

One of the most widely used mouse aortic aneurysmAA () models is the subcutaneous infusion of angiotensin II into *Apoe*
^–/–^ or C57BL/6J mice through osmotic pumps, which is a reliable and reproducible technique to induce both TAA and AAA [8]. This mouse model system has been used to study the candidate genes by ameliorating or enhancing angiotensin II-induced AAAs [9,10,11]. This study shows that *Loxl4* abrogation did not exaggerate angiotensin II-induced thoracic or abdominal aortic aneurysm in mice by using the angiotensin II-induced TAA or AAA mouse model system.

## 2. Materials and Methods

### 2.1. Patients and Whole-Exome Sequencing

A total of 219 unrelated Chinese Han patients with TAAD, who were evaluated at Beijing Anzhen Hospital, Capital Medical University, were recruited from 2013 to 2018. No patients had a family history of TAAD or evidence of a syndromic form of TAAD. Written informed consent was obtained from each of the subjects participating in this study according to a research protocol approved by the Ethical Review Board of Beijing Children’s Hospital, Capital Medical University (Protocol ID: 2020-k-52, approved in March 2020). All mice in this study were bred in the animal facility of the Institute of Genetics and Developmental Biology (IGDB), Chinese Academy of Sciences. All procedures were approved by the Institutional Animal Care and Use Committee of IGDB (Protocol ID: KYD-2018-005, approved in March 2018). The study followed the Declaration of Helsinki.

Whole-exome sequencing was performed, and variants in *LOXL1*, *LOXL2*, *LOXL3*, and *LOXL4* were identified if they had a minor allele frequency ≤ 0.1% in three SNP databases (dbSNP, gnomAD, and 1000 Genomes Project Database). Missense variants were predicted to be damaging or deleterious by at least 2 of 3 functional prediction programs (SIFT, Polyphen-2, and MutationTaster), and having a combined annotation dependent depletion (CADD) score greater than 20 was selected (Table 1). Loss-of-function (LoF) variants, such as stop gain, stop loss, frameshift indels, and splice site variants (2 nt plus/minus the exon boundary), were considered to be deleterious.

### 2.2. Cluster and Homology Analysis

The protein sequences were from NCBI-Protein (https://www.ncbi.nlm.nih.gov/protein/, accessed on 8 February 2021). Amino acid sequences were aligned by ClustalW MEGA 7.0 [12]. The alignment diagram was drawn with GeneDoc. The evolutionary tree was constructed using the Neighbor-Joining method with Mega 7.0 [13]. The domain structures of the LOX family were analyzed by SMART (http://smart.embl-heidelberg.de/, accessed on 8 February 2021) and UCSC (https://www.genome.ucsc.edu/, accessed on 8 February 2021). Domain identity among human and mouse LOXL4 was computed with NCBI-BLAST (https://blast.ncbi.nlm.nih.gov/Blast.cgi, accessed on 8 February 2021).

### 2.3. Loxl4 Knockout Mice

The scheme of *Loxl4* knockout (KO) mice was designed and constructed by Biocytogen (Beijing, China) using the CRISPR/Cas9 technology. Exons 2~14 were deleted, which was confirmed by genomic PCR and Western blotting. All mice were maintained with C57BL/6J genetic backgrounds. Experiments on mice were performed according to institutional guidelines for laboratory animals.

### 2.4. Isolation and Culture of Mouse Heart Endothelial Cells

The heart organs were dissected from wild-type or *Loxl4*-KO mice and washed with HBSS (Thermo Fisher Scientific, Waltham, MA, USA). The organs were cut into pieces and immersed with 25 mL collagenase II (2 mg/mL) (Thermo Fisher Scientific) at 37 °C by rotating the tube slowly for 45 min. The supernatant was filtered with a 70 μm filter and centrifuged at 4 °C, 400 g for 8 min. The precipitate was resuspended with 1 mL basic medium (DMEM: Penicillin-Streptomycin = 100:1, Thermo Fisher Scientific) and 20 μL anti-mouse CD31 antibody (BD Pharmingen, San Jose, CA, USA) coated magnetic beads (Invitrogen, Carlsbad, CA, USA) was added. After incubation for 30 min at 4 °C, the tube was put on the magnetic separator for 1–2 min, and the supernatant was removed. A total of 1 mL of extraction medium (basic medium: fetal bovine serum (FBS) = 4:1, FBS, Thermo Fisher Scientific) was used to resuspend the precipitate. After washing three times, 0.5 mL 0.25% trypsin solution (Thermo Fisher Scientific) was added to the precipitate and digested at 37 °C for 5 min. A total of 0.5 mL extraction medium was added to stop the trypsin digestion. The mixture was centrifuged at 300 g for 10 min. The growth medium (EBM-2: Penicillin-Streptomycin = 100:1, EBM-2, LONZA, Walkersville, MD, USA) was used to resuspend the precipitate. Aliquots of the medium with cells were seeded into culture plates. The growth medium was added into each well, and the plate was put in an incubator with 37 °C, 5% CO_2_ overnight. The purity of the cells was examined by immunostaining using the endothelial cell marker anti-mouse von Willebrand factor (vWF) (1:200, Abcam, Cambridge, U.K.).

### 2.5. Isolation and Culture of Mouse Aorta Smooth Muscle Cells

The aorta was dissected and put into a 35 mm sterile plate with 1 mL DMEM/F12 (serum-free) (Thermo Fisher Scientific). The adventitial tissue was removed under the microscope to clearly expose the blood vessel, which was quickly cut into 1–2 mm size and digested with collagenase II (0.75 mg/mL) for 1 h. A total of 1 mL DMEM/F12 with 20% FBS was added to terminate digestion. After centrifugation at 1000 rpm for 3 min, the supernatant was discarded, and 1 mL of fresh medium (80% DMEM/F12, 20% FBS, 1% Penicillin-Streptomycin) was added to resuspend the cells. The cells were seeded in a 35 mm dish that was pre-paved with gelatin. After 3 days in culture, the purity of the cells was examined by immunostaining using smooth muscle cell (SMC) marker anti-mouse α-SMA antibody (1:200, Sigma-Aldrich, Saint Louis, MO, USA).

### 2.6. DNA Extraction and Genotyping

Mouse toes or tails were cut and lysed by 500 µl lysis buffer (Tris-HCl 100 mM (pH 8.0), EDTA-2Na 5 mM (pH 8.0), NaCl 200 mM, 0.2% SDS, and 0.1 mg/mL proteinase K) at 55 °C for 3 h. Then 300 µl 5 M NaCl was added and mixed thoroughly. The mixture was centrifuged at 12,000 rpm for 15 min, and 400 μL supernatant was collected. An equal volume of ethanol was added and mixed thoroughly. The mixture was centrifuged at 12,000 rpm for 10 min, and then the supernatant was discarded. The precipitate was washed twice with 75% ethanol. After the liquid was evaporated, 50 μL of double-distilled H_2_O was added to dissolve the DNA. A forward primer located on the upstream of the deleted region, a reverse primer located on the deleted region, and another reverse primer located on the downstream of the deleted region were used for genotyping PCR assays (Figure 1A). The primer’s sequences were: F1: 5′-TAATGGAGGCTGACCTTGGACACTTG-3′, R1: 5′-GCCATAACACCACTGTCTGGCTTCT-3′, and

R2: 5′-GATGAGCCTTTGGGTTGAGATTTCCCCA-3′, respectively. The amplified products are shown in Figure 1B.

### 2.7. Western Blotting

The cultured heart ECs and aorta SMCs from wild-type and *Loxl4*-KO mice were collected. Total proteins were extracted from these cells. Protein lysate was separated with 8% SDS-PAGE gels and transferred to PVDF membranes (Millipore, Burlington, MA, USA). The membrane was blocked with 5% skim milk (BD, San Jose, CA, USA) for 1 h at room temperature and then incubated with the indicated primary antibodies of LOXL4 (1:200, Santa Cruz, Dallas, TX, USA), β-actin (1:20000, Sigma-Aldrich) at 4 °C overnight. After incubation with horseradish peroxidase (HRP)-conjugated secondary antibodies (ZSGB-Bio, Beijing, China) for 1 h at room temperature, the membrane was exposed with chemiluminescence apparatus (Beijing Sage Creation, Beijing, China) and visualized by ECL (GE, Boston, MA, USA).

For determining the serum metalloproteinase-9 (MMP9) level, mouse serum was diluted 1:10 with PBS (Hyclone, Logan, UT, USA). An equal volume of diluted samples was separated with 8% SDS-PAGE gel. The membrane was incubated with the primary antibody of MMP9 (1:200, Santa Cruz). The other steps are the same as above.

### 2.8. *LOX*/*LOXL* Activity

The heart and aorta tissues from 3-month-old WT and *Loxl4*-KO mice were homogenized in protein extraction buffer (6 M urea, 10 mM Tris PH 7.4 with protease inhibitor) using multi-sample tissue lyser (Jingxin, Shanghai, China) at 60 HZ/s for 60 s and centrifuged at 15,000× *g* for 15 min at 4 °C. Then the lysed tissue was incubated with the LOX substrate (Abcam) for 30 min at 37 °C. The fluorescence from released hydrogen peroxide in horseradish peroxidase (HRP)-coupled reactions was measured using a fluorescence microplate reader with excitation and emission wavelengths at 535 and 587 nm.

### 2.9. Metalloproteinase-2 (MMP2) Activity

Peripheral blood was collected from the hearts of 12-month-old mice. Blood samples were incubated at 37 °C for 30 min, centrifuged at 4 °C, 4000 rpm for 5 min, and then the supernatant was collected. The serum MMP2 activity was measured by an ELISA kit (Abcam) according to the manufacturer’s instruction.

### 2.10. Real-Time PCR

Total RNA from the whole aorta of 12-month-old mice was extracted using RNeasy Mini Kit (QIAGEN, Hilden, Germany). A total of 2 µg RNA was synthesized to cDNA using the Bio-Rad transcript cDNA synthesis kit (Bio-Rad, Hercules, CA, USA). The cDNA was used as templates to amplify Mmp9 and Gapdh. Primers of Mmp9 were 5′- GGACCCGAAGCGGACATTG-3′ and 5′-CGTCGTCGAAATGGGCATCT-3′. Primers of Gapdh were 5′-AGGTCGGTGTGAACGGATTTG-3′ and 5′-GGGGTCGTTGATGGCAACA-3′. Real-time PCR was performed using SYBR Green (TIANGEN, Beijing, China) in the Rotor-Gene Q system (QIAGEN).

### 2.11. Implantation of Mini-Pumps

To induce AA formation, mice of 14 weeks or 20 weeks were selected to be surgically implanted with ALZET osmotic mini-pumps (Model 1004, Durect Corporation, Cupertino, CA, USA) to infuse angiotensin II (Sigma-Aldrich) (1.44 mg/kg/d or 2.16 mg/kg/d) for 28 days, as described previously [14]. In our study, all experiments were conducted using male mice. Controls were age-matched littermates. Four weeks after infusion, mice were euthanized. Aortas were harvested, cleaned, and imaged. Thoracic and abdominal aortas were histochemically stained and evaluated.

### 2.12. Tail Cuff Blood Pressure Measurements

Bp-2010A (Softron, Beijing, China) was used to measure the tail artery blood pressure of mice. In order to eliminate the experimental error caused by nervous discomfort, baseline measurements were performed after 6 days of training before the implantation of mini-pumps. Blood pressure was measured on the seventh, fourteenth, twenty-first, and twenty-eighth day after angiotensin II infusion at a fixed time (8:00 to 12:00 in the morning). Each experimental mouse was tested 10 times for training adaptation, and the formal measurement was repeated 10–15 times. An average of the measured pressure represents the recorded mouse blood pressure.

### 2.13. Aortic Ultrasonography and Measurement of Aortic Diameters

After 28 days of angiotensin II infusion, mice were anesthetized with 1% isoflurane, and their thorax and abdomen were depilated and coated with a developer. Vevo 2100 Imaging System (FUJIFILM VisualSonics, Toronto, QC, Canada) was used, which is equipped with a 30-MHz transducer to perform ultrasound scans in B-mode, including the longitudinal images of ascending aorta, aortic arch, and abdominal aorta then measured the internal diameter of the aorta. The branch of the cephalic trunk and up the branch of the coeliac trunk about 1 cm was selected as the anatomy mark of the diameter measurement of the aortic arch and abdominal aorta, respectively. AAA could be diagnosed when the ratio of the diameter of the suprarenal aorta and the undilated adjacent aorta is ≥1.5 [15].

### 2.14. Pulse Wave Velocity (PWV) Measurement in Mice

In order to evaluate the arterial stiffness in mice aorta, mice PWV were measured as described previously [16]. In brief, mice were anesthetized with 2% pentobarbital sodium and placed on a temperature-controlled electrocardiogram board. Doppler spectrograms of aortic flow at the branch of the left carotid artery and coeliac trunk were acquired with a 30-MHz pulsed Doppler probe (FUJIFILM VisualSonics). The propagation time for the pulse wave moving from the branch of the left carotid artery to the coeliac trunk was measured. Mice were sacrificed in order to evaluate the distance between the branch of the left carotid artery and the coeliac trunk. Then aortic PWV was calculated by the distance between two measurement locations divided by the propagation time.

### 2.15. Histology and Immunohistochemistry

Mouse aortic tissue was fixed in 4% paraformaldehyde, adventitial tissue was carefully removed, the tissue was laid out on a black gel plate, and an image of the aorta was recorded. Aortic was embedded in paraffin and sectioned. Aortic sections (5 µm) were treated with Masson’s trichrome staining, elastic staining, and hematoxylin and eosin (HE) staining, respectively, following routine procedures. Images were captured by the Nikon Microscope Eclipse H550S (Nikon, Tokyo, Japan).

### 2.16. Statistics

Data were presented as mean ± standard error of the mean (SEM). Student’s *t*-test was used to compare different groups. All statistical tests were two-sided, and a significant difference was considered when the *p*-value was less than 0.05. Comparisons of the aortic aneurysm incidence rates were analyzed by the method of Fisher’s exact test with a statistical threshold of 0.05. GraphPad Prism version 7.00 (San Diego, CA, USA) was used for statistical analyses.

## 3. Results

### 3.1. Variants of LOXL4 Identified by Whole-Exome Sequencing

In order to identify new candidate genes related to TAAD, we pinpointed seven variants from eight patients in the *LOXL4* gene of the *LOX* gene family (Table 1). The LOX family is essential to the biogenesis of connective tissue, which catalyzes the first step in the formation of crosslinks in collagens and elastin. Six of these variants are missense mutations, and one is a frameshift mutation. All these variants are heterozygous mutations and classified as VUS except for the frameshift variant c.1588delG, which was predicted to result in premature truncation of the protein and classified as likely pathogenic by following the American College of Medical Genetics and Genomics (ACMG) guidelines [17]. Inferred from the *LOX* gene for TAAD, multiple variants from different patients in the *LOXL4* gene highlighted that the *LOXL4* gene could be another causative gene of TAAD.

### 3.2. Cluster and Homology Analysis of Human and Mouse *LOX* Family

To ascertain the mouse LOXL4 homolog of the human LOX family, homology analysis between mouse and human LOX family was conducted. The evolutionary tree of the LOX family showed that the LOX family converged within humans and mice (Figure 1A). Structure analyses of the mouse LOX family by SMART and UCSC showed that they all contain a highly conserved carboxyl terminus that comprises a copper-binding motif and a lysyl-tyrosyl-quinone (LTQ) cofactor, which are required for protein conformation and lysyl oxidase catalytic activity, respectively. The C terminus also contains a cytokine receptor-like (CRL) domain. The N- termini of the LOX proteins are more divergent, with LOXL2, LOXL3, and LOXL4 containing four scavenger receptor cysteine-rich (SRCR) domains that are thought to be involved in cell adhesion and protein-protein interaction [18,19] (Figure 1B). Sequence alignment results showed that mouse LOXL4 shared 87.1% identity with human LOXL4 (Figure 1C) and that mouse LOXL4 lysyl oxidase (LO) domain shared 51.24% identity with mouse LOX LO domain (Figure 1D). These results showed the LOX family members are conserved between humans and mice, suggesting that LOX activity may be redundant in these species.

**Figure 1 genes-12-00513-f001:**
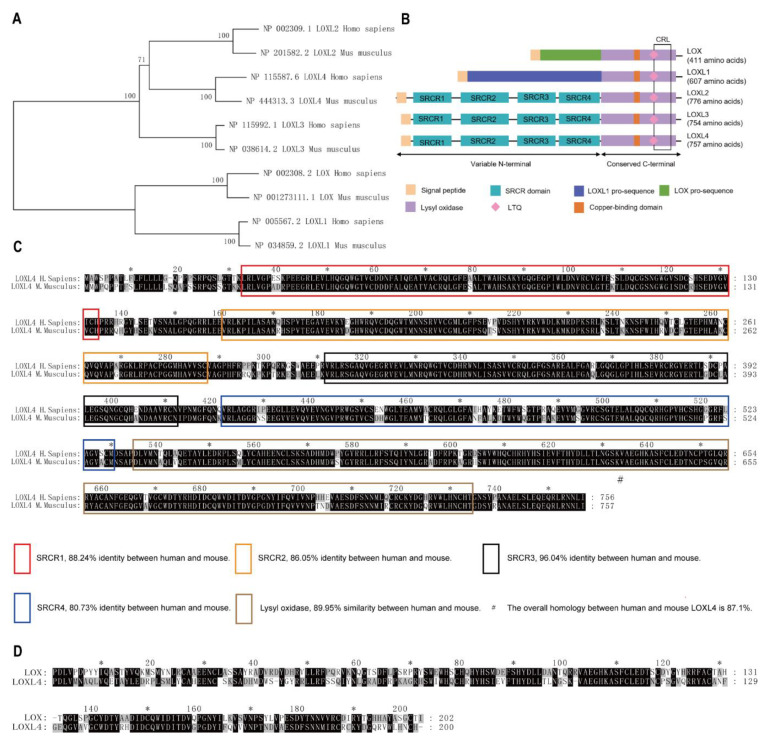
Cluster and homology analysis of human and mouse LOX families. Homology analysis of LOX families between humans and mice. (**A**) The evolutionary tree of the LOX family in humans and mice. The percentage of replicate trees in which the associated taxa clustered together in the bootstrap test (1000 replicates) was shown next to the branches. (**B**) Structure analyses of mouse LOX family members. LTQ, lysyl-tyrosyl-quinone; CRL, cytokine receptor-like. SRCR, scavenger receptor cysteine-rich. (**C**) The sequence alignment of mouse LOXL4 and human LOXL4. (**D**) The sequence alignment of mouse LOXL4 lysyl oxidase (LO) domain and mouse LOX LO domain. *, indicates the number of amino acids, which is marked every twenty amino acids.

### 3.3. Loxl4-KO Mice Do Not Attenuate Lysyl Oxidase Enzyme Activity

To study whether loss-of-function of the *Loxl4* gene in mice develops TAAD as observed in human patients, we generated a *Loxl4*-KO mouse for in vivo assays. The exons from 2 to 14 of *Loxl4* were deleted using the CRISPR/Cas9 system on C57BL/6 background mice (Figure 2A). Deletion of the *Loxl4*-KO was confirmed by genomic PCR (Figure 2B) and Western blotting in aorta smooth muscle cells and heart endothelial cells (Figure 2E,F). The purity of these cells was confirmed by indicative cell-type markers (Figure 2C,D). These results demonstrated that LOXL4 was detectable in the wild-type aorta smooth muscle cells and heart endothelial cells which were depleted in the *Loxl4*-KO mice (Figure 2E,F).

The lysyl oxidase activity of LOX/LOXL can prevent the occurrence of AA in mice. Loss of lysyl oxidase activity will seriously affect the stability of the aortic extracellular matrix. We evaluated whether *Loxl4* deletion will compromise the activity of lysyl oxidase. We found that *Loxl4*-KO mice did not attenuate lysyl oxidase enzyme activity in the heart and aorta tissues (Figure 2G). In addition, we did not observe higher gene expression of other LOX family members by real-time PCR in *Loxl4*-KO mice compared with wild-type mice, suggesting that other existing LOX members may be sufficient for lysyl oxidase activity to compensate for the deficiency of LOXL4.

### 3.4. Loxl4-KO Mice Do Not Spontaneously Develop Aortic Aneurysm at the Age of 12 Months

To study whether *Loxl4* deletion mice spontaneously develop aortic aneurysm, wild-type and *Loxl4*-KO mice were housed in a quiet room at 25 °C with a 12-h light/dark cycle and free access to food and water until the age of 12 months. Then these mice underwent blood pressure measurement, aortic ultrasonography monitoring, and measurement of aortic diameters. There was no difference in aortic diameters between wild-type and *Loxl4*-KO mice (Figure 3A–C). The *Loxl4*-KO mice did not have significantly different systolic blood pressure (wild-type: 119.6 ± 3.32 vs. *Loxl4*-KO: 112.4 ± 3.042 mmHg; *p* = 0.1522) (Figure 3D). Peripheral blood was also collected from mouse hearts, and the serum MMP2 activity was measured by ELISA. There was no difference between wild-type and *Loxl4*-KO mice (Figure 3E). It has been reported that a high plasma level of MMP9 serves as an important biological marker that is indicative of the presence of AAA [20]. Real-time PCR and Western blotting were conducted to examine the expression of MMP9 in whole aorta or serum, respectively. Our results showed that there was no difference in MMP9 expression between wide-type and *Loxl4*-KO mice (Figure 3F,G). These results suggest that *Loxl4*-KO mice do not spontaneously develop AA at the age of 12 months.

### 3.5. No Indication of Angiotensin II-induced Aortic Aneurysm in Wild-Type and Loxl4-KO Mice at the Age of 14 Weeks

To study whether administration of angiotensin II can induce AA formation, 14-week-old wild-type and *Loxl4*-KO mice were selected to implant with ALZET osmotic mini-pumps infusing angiotensin II at a concentration of 2.16 mg/kg/d for 28 days. However, there was no AA occurrence both in wild-type or *Loxl4*-KO mice (Figure 4A,B). In order to evaluate the arterial stiffness in the mice model, aorta PWV was measured. There was no difference between the two groups (Figure 4C). Furthermore, there was no aortic diameter difference between wild-type and *Loxl4*-KO mice (Figure 4D). In addition, there was no significant difference in maximum aortic diameter between wild-type and *Loxl4*-KO mice exposed to saline or angiotensin II (Table 2). Together, these results suggest that both wild-type and *Loxl4*-KO mice do not have angiotensin II-induced AA at the age of 14 weeks.

### 3.6. Loxl-KO Does Not Exaggerate Angiotensin II-Induced TAA or AAA in Mice at the Age of 20 Weeks

To further study whether the development of angiotensin II-induced AA is age-dependent, 20-weeks old wild-type and *Loxl4*-KO mice were selected to implant with ALZET osmotic mini-pumps infusing angiotensin II at a concentration of 1.44 mg/kg/d for 28 days (*n* =23 and 24, respectively). Two wild-type mice and one *Loxl4*^−/−^ mouse were subjected to sudden death in the first 7 days. In the surviving mice, systematic analysis of aortic ultrasonography, the recorded image of the aorta and aortic sections revealed aortic aneurysm in 6 of 21 wild-type mice (28.6%), and in 7 of 23 *Loxl4*-KO mice (30.4%) (*p* = 0.8924; Figure 5A,B). The difference in the occurrence of aortic dissections in wild-type versus *Loxl4*-KO mice was not statistically significant (Figure 5C). Angiotensin II-induced blood pressure elevations were detected in both wild-type and *Loxl4*-KO mice, and these elevations were not different between the two groups (Figure 5D). Aorta PWV was also measured to evaluate the arterial stiffness, and there was no difference between these two groups (Figure 5E). There was no aortic diameter difference between wild-type and *Loxl4*-KO mice (Figure 5F). Likewise, there was no significant difference in maximum aortic diameter between wild-type and *Loxl4*-KO mice exposed to saline or angiotensin II (Table 2). These results suggest that deficiency of mouse LOXL4 does not result in AA, indicating that human *LOXL4* is not a causative gene of AA.

## 4. Discussion

There are about 30 known genes associated with TAAD, most of which are related to the TGF-β signaling pathway or smooth muscle cell contraction [21]. However, the inherited forms of TAAD are largely unrecognized, and the genes found at present can only explain a proportion of the occurrence of TAAD. One of the important functions of the LOX family is to catalyze the cross-linking of lysine and hydroxylysine between collagen fibers and elastic fibers, which play an important role in the regulation of the extracellular matrix [22]. It has been reported that in the 6–8 weeks of *Apoe*^−/−^ mice, which accepted two different treatments: the first group was treated with a high-fat diet and pump embedding sustained release of angiotensin II (0.75 mg/kg/day) for 4 weeks; and the second group was treated with a high-fat diet and pump embedding sustained release of angiotensin II (0.75 mg/kg/day) and a LOX inhibitor β-aminopropionitrile (BAPN, 100 mg/kg/day) for 4 weeks. The incidence of AAA in the second group was much higher than that in the first group (90% vs. 15%) [23]. These results suggest that the lysyl oxidase activity of LOX/LOXL plays an important role in the cross-linking of collagen and elastic fibers in the aorta. This function is essential to maintain the elasticity and integrity of various human tissues. By analyzing the WES data from 219 sporadic TAAD patients, we found seven variants in the *LOXL4* gene. We reasoned that the *LOXL4* gene could be another candidate gene of TAAD from the *LOX* gene family.

*LOX* gene homozygous KO mice died during pregnancy or postnatal stage, and autopsy showed large aortic aneurysms [4]. Compared with the wild-type mice, the aortic wall of *LOX*
*^−/−^* mice was significantly thicker, and the aortic lumen was significantly smaller. Interestingly, the mRNA expression of *Loxl1~4* in *Lox*
^−/−^ mice also decreased [24], which indicates that there is a regulatory interaction among the members of the LOX family. Compared with wild-type mice, the activity of lysyl oxidase in the aorta and lung of *Lox*
^−/−^ mice decreased by 60% [25]. This suggests that there were other members of this family expressing lysyl oxidase activity in these two tissues. However, in our study, we found that unlike the *Lox*
^−/−^ mice, *Loxl4*^−/−^ mice did not attenuate lysyl oxidase enzyme activity in the mouse tissues of the heart and aorta, and the *Loxl4*^−/−^ mice did not spontaneously develop AA at the age of 12 months in our study. Cluster and homology analysis indicated that LOX family members converged within humans and mice. Mouse LOXL4 shared 87.1% identity with human LOXL4, and the mouse LOXL4 LO domain shared 51.24% identity with the mouse LOX LO domain. Our results suggest that LOXL4 may not be the major player of the lysyl oxidase enzyme activity in the cardiovascular tissues and that the *Lox* gene and other *Loxl* genes may maintain the enzyme activity in the *Loxl4*^−/−^ mice.

The animal model system of subcutaneous infusion of angiotensin II has been used to recapitulate many features of AAA observed in humans. Male mice are more vulnerable to have induced AAA than female mice [26] because of the influence of androgen [27,28]. Angiotensin II infusion into mice can lead to a profound expansion of the thoracic aortic region, which is predominantly restricted to the ascending aorta. This region is also the most common region for TAA in humans [29], but the incidence of induced TAA is lower than AAA in this mouse model system. In our study, angiotensin II-induced blood pressure elevations were detected in both wild-type and *Loxl4*^−/−^ mice, but these elevations were not different between the two groups. In other words, we found that *Loxl4*^−/−^ mice did not induce a more severe thoracic or abdominal aortic aneurysm. In conclusion, deletion of *Loxl4* did not involve the underlying pathogenesis of angiotensin II-induced AA. Further study is needed to explore whether *LOXL4* plays an addictive role in the development of AA when other *LOX* gene family members are deficient.

## Figures and Tables

**Figure 2 genes-12-00513-f002:**
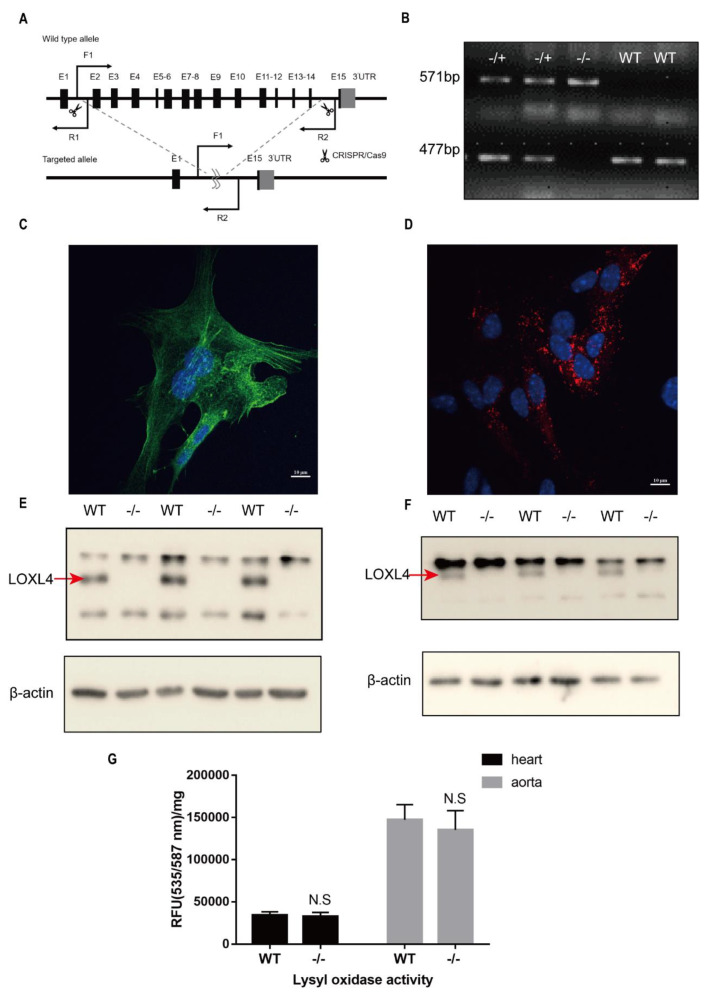
Verification of the deletion of *Loxl4* in mice and measurements of the lysyl oxidase activity of hearts and aortas. (**A**) The scheme of *Loxl4* knockout strategy. Exons 2~14 were targeted to be deleted by CRISPR/Cas9 technology. F1, R1, and R2 indicate the locations of the primers used for genotyping. (**B**) Verification of *Loxl4* deletion from genomic level. The lower bands were amplified with primers F1 and R1 (477 bp). These bands can only be amplified in wild-type or heterozygous mice and cannot be amplified in homozygous mice because of the absence of the sequence of primer R1. The upper bands were amplified with primers F1 and R2 (571 bp). These bands can only be amplified in heterozygous and homozygous mice, for the sequence between F1 and R2 is too long to be amplified in wild-type mice. (**C**) Immunostaining of actin (green) on aorta smooth muscle cells and (**D**) von Willebrand factor (Vwf) (red) on heart endothelial cells. Pictures were taken at a magnification of 100×. The bar is 10 µm. (**E**) Verification of *Loxl4* deletion at the protein level in aorta smooth muscle cells and (**F**) heart endothelial cells. An about 70 kDa specific LOXL4 band was detected in wide-type mice but not in homozygous mice. (**G**) Lysyl oxidase activity of wild-type (*n* = 6) and *Loxl4* knockout (KO) mice (*n* = 7) was evaluated by fluorescence (RFU) values per protein quantity (mg). *p* = 0.5354 and 0.3165 in hearts and aortas, respectively. WT, wild-type; −/+, heterozygote; −/−, KO; N.S, no significance.

**Figure 3 genes-12-00513-f003:**
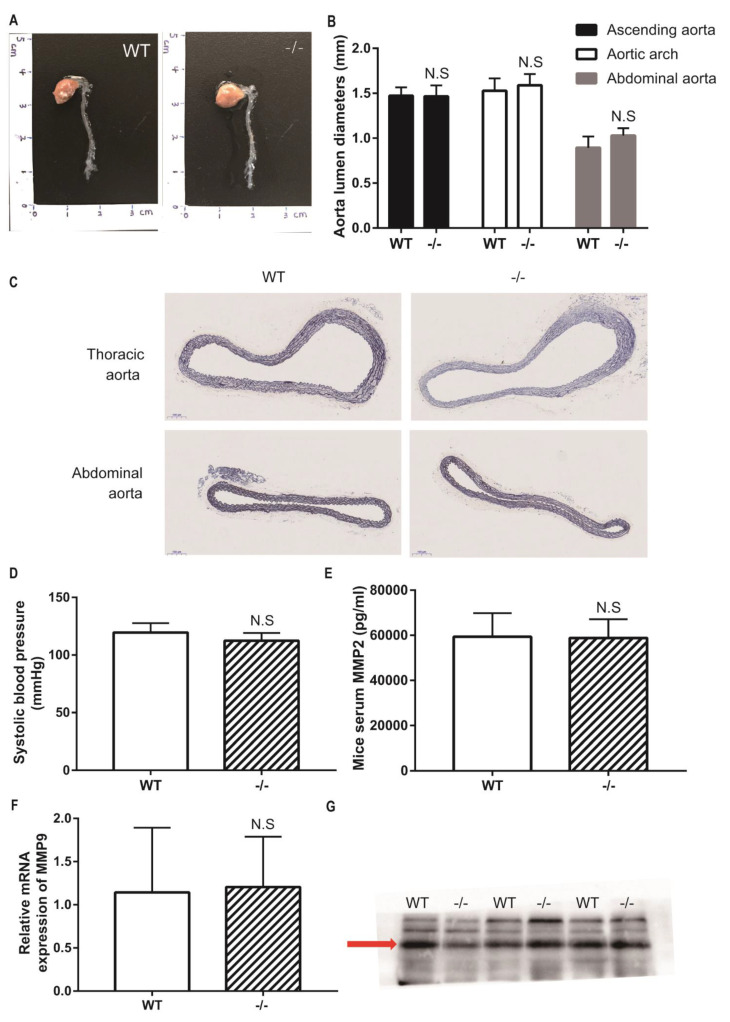
*Loxl4*-KO mice did not spontaneously develop aortic aneurysm at 12 months. (**A**) Representative aortic images from wild-type and *Loxl4*-KO mice at the age of 12 months. (**B**) Lumen diameters of ascending aorta, aortic arch, and abdominal aorta from wild-type (*n* = 5) and *Loxl4*-KO mice (*n* = 5), *p* = 0.9356, 0.4829 and 0.0770, respectively. (**C**) Mouse thoracic aortas and abdominal aortas were sectioned and stained by elastic staining. Pictures were taken at a magnification of 10×. The bar is 100 µm. (**D**) Blood pressure in wild-type (*n* = 6) and *Loxl4*-KO mice (*n* = 5), *p* = 0.1522. (**E**) Serum Metalloproteinase-2 (MMP2) activity was measured from wild-type (*n* = 15) and *Loxl4*-KO mice (*n* = 13), *p* = 0.8863. (**F**) Relative mRNA expression of *Mmp9* in whole aortas of wild-type (*n* = 4) and *Loxl4*-KO mice (*n* = 3), *p* = 0.9102. (**G**) The serum *Mmp9* level was measured by Western blotting, and an about 100 kDa specific metalloproteinase-9 (MMP9) band was detected in wide-type and *Loxl4*-KO mice, *p* = 0.6533.

**Figure 4 genes-12-00513-f004:**
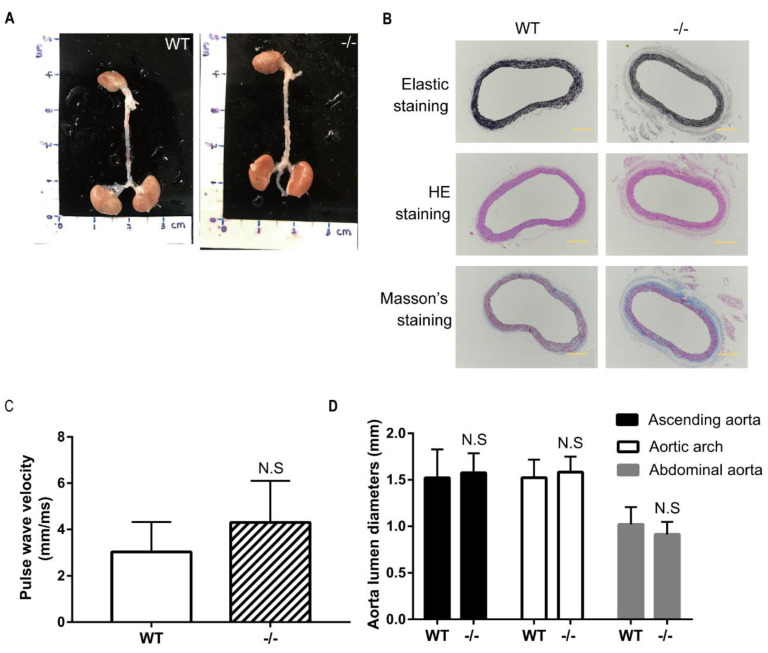
No indication of angiotensin II-induced aortic aneurysm in wild-type and *Loxl4*-KO mice at the age of 14 weeks. Angiotensin II (2.16 mg/kg/d) was infused in wild-type or *Loxl4*-KO mice at the age of 14 weeks. (**A**) Representative aortic images from wild-type and *Loxl4*-KO mice after angiotensin II infusion. (**B**) Masson’s staining, hematoxylin and eosin (HE) staining, and elastic staining of thoracic aortas from wild-type and *Loxl4*-KO mice. Pictures were taken at a magnification of 10×. The bar is 200 µm. (**C**) Aortic pulse wave velocity (PWV) of mice receiving angiotensin II infusion for 4 weeks, *p* = 0.1877. (**D**) Lumen diameters of ascending aorta, aortic arch, and abdominal aorta from wild-type (*n* = 16) and *Loxl4*-KO mice (*n* = 11), *p* = 0.6051, 0.4197, and 0.1669, respectively.

**Figure 5 genes-12-00513-f005:**
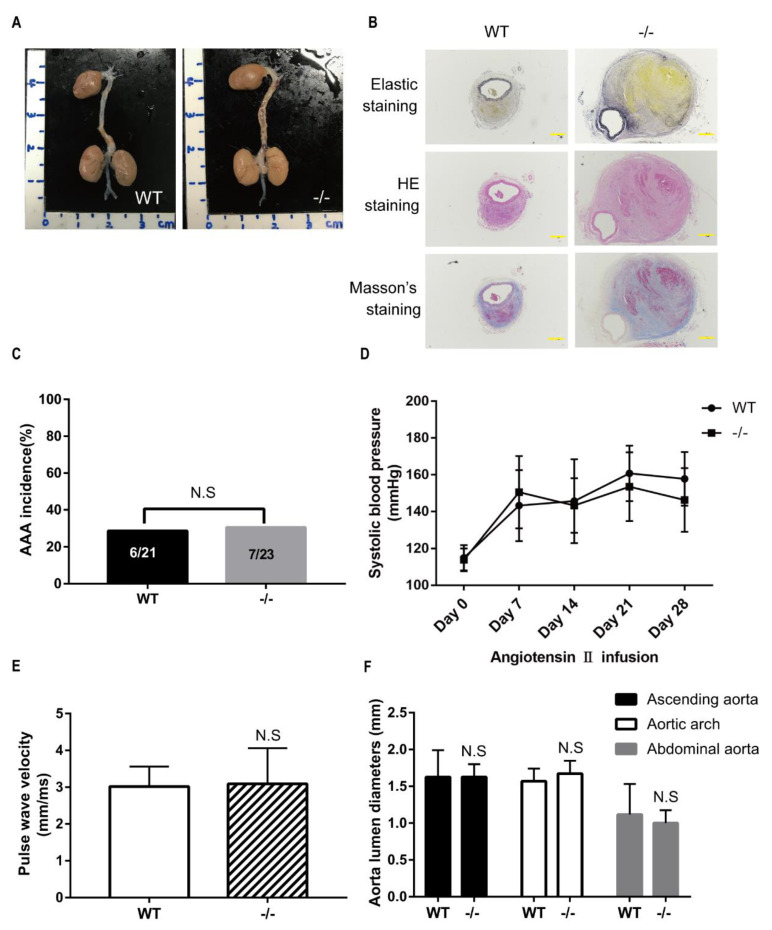
*Loxl4* deletion did not exaggerate angiotensin II-induced thoracic aortic aneurysms (TAA) or abdominal aortic aneurysms (AAA) in mice at the age of 20 weeks. (**A**) Representative aortic images from wild-type and *Loxl4*-KO mice after angiotensin II (1.44 mg/kg/d) infusion. (**B**) Masson’s staining, HE staining, and elastic staining of the abdominal aortic aneurysm from wild-type and *Loxl4*-KO mice. Pictures were taken at a magnification of 4×. The bar is 400 µm. (**C**) Aortic aneurysm was detected in 6 of 21 wild-type mice and in 7 of 23 *Loxl4*-KO mice, *p* = 0.8924. (**D**) Blood pressure changes in angiotensin II-infused wild-type and *Loxl4*-KO mice. Days of measurement are indicated with respect to pump implantation. Day 0 = basal blood pressure before pump implantation. (**E**) Aortic pulse wave velocity (PWV) of mice received angiotensin II infusion for 4 weeks, *p* = 0.8734. (**F**) Lumen diameters of ascending aorta, aortic arch, and abdominal aorta from wild-type (*n* = 18) and *Loxl4*-KO mice (*n* = 13), *p* = 0.9842, 0.1145, and 0.3418, respectively.

**Table 1 genes-12-00513-t001:** Variants of *Loxl4* gene in 8 unrelated patients with thoracic aortic aneurysm and dissection (TAAD).

NT Change	AA Change	Domain	Allele Frequency		Functional Prediction Program				ACMG
			1000 Genomes	gnomAD_exome	SIFT	PP2	Mutation Taster	CADD	
c.G1042A *	p.V348M	SRCR3	0.0008	0.0005	D	D	D	28.6	VUS
c.G1144A	p.G382R	SRCR3	-	4.07 × 10^−6^	D	D	D	34	VUS
c.G1291A	p.E431K	SRCR4	-	2.033 × 10^−5^	D	B	D	31	VUS
c.1588delG	p.D530fs	-	-	-	-	-	-	-	LP
c.C1804T	p.R602C	-	0.0004	4.873 × 10^−5^	D	D	D	34	VUS
c.G1805A	p.R602H	-	-	5.279 × 10^−5^	D	D	D	27.4	VUS
c.G1897A	p.V633M	-	-	-	D	D	D	33	VUS

Note: NT, nucleotide; AA, amino acid; SRCR: scavenger receptor cysteine-rich; SIFT (D: damaging); PP2, polyphen-2 (D: damaging); Mutation Taster (D: disease-causing); CADD, combined annotation dependent depletion; ACMG, American College of Medical Genetics and Genomics guidelines (LP, likely pathogenic; VUS, variant uncertain significance); -, absence. * This variant is from two unrelated patients.

**Table 2 genes-12-00513-t002:** Maximum aortic diameters of wild-type and *LOXL4*-KO mice exposed to saline or angiotensin II.

Group	Diameter (mm)		
	Ascending Aorta	Trans Arch	Abdominal Aorta
NC, 14 weeks old mice (WT, n = 4)	1.533 ± 0.0475	1.542 ± 0.06587	0.8576 ± 0.03747
NC, 14 weeks old mice (KO, n = 4)	1.488 ± 0.03497	1.584 ± 0.05354	0.860 ± 0.08984
Angiotensin II (2.16 mg/kg/day), 14-week-old mice (WT, n = 16)	1.521 ± 0.07614	1.523 ± 0.04885	1.005 ± 0.04551
Angiotensin II (2.16 mg/kg/day), 14-week-old mice (KO, n = 11)	1.577 ± 0.06314	1.582 ± 0.05017	0.9133 ± 0.04048
Angiotensin II (1.44 mg/kg/day), 20-week-old mice (WT, n = 18)	1.627 ± 0.08559	1.570 ± 0.04004	1.119 ± 0.09769
Angiotensin II (1.44 mg/kg/day), 20-week-old mice (KO, n = 13)	1.629 ± 0.04792	1.672 ± 0.04852	1.000 ± 0.04899

## Data Availability

The data presented in this study are available on request from the corresponding author. The data are not publicly available due to privacy or ethics.

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
