# Peer review of "LOXL4 Abrogation Does Not Exaggerate Angiotensin II-Induced Thoracic or Abdominal Aortic Aneurysm in Mice"

_genes, 2021, doi:10.3390/genes12040513_

Round 1
Reviewer 1 Report
Thank you for the elegant manner in which this study was conducted and the clear manner in which it was presented.
Do you think there was a defect in your Knock-Out Angiotensin model that interfered with manifestation of thoracic aortic changes. This is a complex model. Have you used it before?
How do you reconcile the apparent clinical importance of your LOX variants in patients with the failure to demonstrate an effect in your mouse model?
Do you plan further work on this issue, or are you done?
Reviewer 2 Report
- It would be interesting to look at compensation of other Lox family members in the Lox4 knock-out animals. It could be possible the effects you are seeing are a result of higher levels of the other family members in the total knock-out mice. It would be recommended to suggest that cell specific-knock out animals should be considered before completely ruling out effects of Lox4 on aneurysms.
- It would also be interesting to conduct further histological analysis to determine if the Lox4 knock-out animals cause some other sort of changes to the aorta during aneurysm formation.
- It would be of interest to examine the histology in the different locations measured in the diameter measurements. Could there be changes that are missed because of not assessing the other locations histologically?
- This zoomed out images of the whole aorta are difficult to assess. Can there also be images of the aorta zoomed in to see that there are in fact aneurysms in the aorta?
Reviewer 3 Report
The work is interesting and the statistical analysis supports the conclusions reported.
Section 2.9 and 3.4 mentions MMP2 metalloproteases. it could also be useful to add the value of the MMP9 (page 1 Introduction)
EVAR and OPEN treatment of abdominal aortic aneurysm: What is the role of MMP-9 in the follow-up?
Ascoli Marchetti A, Pratesi G, Di Giulio L, Battistini M, Massoud R, Ippoliti A. J Med Vasc. 2017 Feb;42(1):21-28. doi: 10.1016/j.jdmv.2017.01.004. Epub 2017 Apr 18. PMID: 28705444
Round 2
Reviewer 2 Report
Thank you for your revisions. No additional comments.